# Seasonal Body Composition Changes in Elite Rugby Players: DXA and Anthropometry-Based Comparison of Backs and Forwards

**DOI:** 10.3390/jfmk10030357

**Published:** 2025-09-18

**Authors:** Blanca Couce, Anel E. Recarey-Rodríguez, Selene Baos, Helios Pareja-Galeano, María Martínez-Ferrán

**Affiliations:** 1 Facultad de Ciencias de la Salud, Universidad Isabel I, 09003 Burgos, Spain; blanca.couce@ui1.es (B.C.); aneleduardo.recarey@ui1.es (A.E.R.-R.); selene.baos@ui1.es (S.B.); 2Department of Physical Education, Sport and Human Movement, Universidad Autónoma de Madrid, 28049 Madrid, Spain; 3Faculty of Health Sciences, Universidad de Burgos, 09001 Burgos, Spain; mmferran@ubu.es

**Keywords:** team sports, adiposity, fat-free mass, dual energy x-ray absorptiometry, somatotype

## Abstract

**Background**: Body composition analysis in rugby is necessary for profiling athletes for ideal positioning, the establishment of standards and the development of nutritional and training strategies for improvement. This study aimed to assess the body composition of elite rugby players and examine seasonal variations between backs and forwards using dual-energy X-ray absorptiometry (DXA) and anthropometry. **Methods**: Thirty-two rugby players (25.97 ± 4.51 years; 93.00 ± 15.39 kg; 181.77 ± 6.27 cm) from First Spanish National league team had their body composition assessed using DXA and anthropometry before and after the season. **Results**: The main findings indicated that backs exhibited significant increases in total mass, lean mass and fat-free mass, whereas forwards showed and increased only in bone mass. In terms of somatotype, significant changes were observed only in backs, who demonstrated increased mesomorphy and ectomorphy. Within the forwards, front-row players experienced greater increases in LM and FFM compared to other forwards. **Conclusions**: Rugby players show differences in body composition based on the physical demands of their playing position. These positional differences in body composition are influenced by both training adaptations and genetic predispositions, which ultimately determine the suitability of players for specific roles on the field.

## 1. Introduction

Rugby Union (RU) is a team-based contact sport that intersperses high-intensity efforts such as sprints, mauls, and tackles with low-intensity periods that serve as recovery periods for players [1]. Consequently, rugby players are subjected to extreme physical demands, requiring great development of anthropometric qualities, as well as an optimal nutritional and health status in order to maximize their physical performance [2]. As rugby has become more professional, changes in the game have been observed over the years. Currently, up to eight substitutions can be made per match, so it is not so important that forwards endure 80 min, but rather that they have more power to fight for possession. Backs are also increasingly involved in tasks that were previously the exclusive preserve of the forwards. Therefore, over time, an increase in weight and muscle mass has been observed in both backs and forwards due to the evolution of the game rules [3,4,5].

RU involves large variations in body composition depending on the position played; this is why it is essential to study the positions separately [4]. Unlike other sports, body composition of professional male rugby players is poorly described in the literature. It is essential to know the anthropometric characteristics and ideal body composition of professional players for each playing position, as this lack of information hinders the professionalisation of amateur players and teams, as they have no standards to strive for [6,7]. Seasonal monitoring of body composition is also very useful to check the impact of the different phases (pre-season and regular season), as well as to avoid undesired changes in the off-season due to detraining or abandoning healthy dietary habits during the rest period. Longitudinal quantification of body composition in rugby players is essential to define normative standards that can be used in athlete profiling, e.g., height and weight strongly correlate with performance, particularly in the scrum, where athletes enhance strength [5,6]. As a result, coaches prioritize size as a key factor for forwards’ success [3]. Quantifying body composition of professional players is also interesting for identifying strengths and weaknesses in order to set improvement goals, performance monitoring, and talent detection purposes, very useful for transfer selection [6,8]. Likewise, it is particularly interesting to know the body composition in order to establish nutritional and training intervention strategies that contribute to improving the performance of the players. In short, knowing the body composition of professional players is key to specifying the ideal anthropometric qualities for achieving greater performance in each playing position and, therefore, to optimizing the player selection process [6,7].

Anthropometry is commonly used as a method to assess the body composition of these athletes because of its practicality: it is portable, accessible, and reliable for monitoring progress [9]. In contrast, dual energy x-ray absorptiometry (DXA) is a powerful diagnostic tool for obtaining more complete data on bone and body composition, as it is the gold standard for bone mineral density (BMD) [9,10,11,12], which is particularly important in rugby due to the high number and intensity of collisions. In the same way, skeletal proportions are also relevant as a limiting factor for muscle mass [6]. Nevertheless, DXA is a limited method due to the high cost, so skinfolds are a useful tool for estimating fat mass in a much more accessible way [12]. Somatotype is a helpful analytical tool in athlete profiling as well; it is defined as the quantification of the shape and composition of the human body [13]. Therefore, DXA is not widely available, so it is helpful to also use anthropometry in order to provide body composition data to other clubs so that they can compare their own data in an easy way. Furthermore, the information provided by these two methods is not the same, so they complement each other. So, while DXA offers much greater accuracy for fat mass, fat-free mass, muscle mass, and bone mass, both for the whole body and for body segments, anthropometry can also be used to assess morphological body proportions, as well as providing data that can be used to quickly and regularly assess fat mass, such as skinfold measurements [11,12].

This study aimed to comprehensively assess body composition and somatotype characteristics of elite rugby players from the Spanish First Division, comparing forwards and backs, and to examine positional and seasonal changes in anthropometric and DXA-derived variables between the pre-season and the end of competitive season. We hypothesize that changes in body composition over the course of the season will differ between playing positions.

## 2. Materials and Methods

### 2.1. Participants

Thirty-two (*n* = 32; 25.97 ± 4.51 years) male rugby players who were competing in the First Spanish National League (Division de Honor A) voluntarily agreed to participate in the study after attending an informative session where the researchers detailed the characteristics of the study. No player on the team refused to participate. No distinction was made according to ethnicity or nationality. The study was conducted in accordance with the Declaration of Helsinki [14] and approved by the Ethics Committee of the Universidad Isabel I (CEI-23-03—PI084).

Athletes were divided into two groups according to position (forwards: *n* = 16; backs: *n* = 16) for analysis and comparison. Forwards were composed by front row (*n* = 8), second row (*n* = 2), and back row (*n* = 6), while backs were composed by scrum-half (*n* = 3), inside-backs (*n* = 5), and outside-backs (*n* = 8). All of them trained 4 days a week, with 1 h in the gym with physical training and 1.5–2 h on the field with coaches, plus a 5th match day. Injuries or illnesses were exclusion criteria; however, all participants were free of injury or illness on test day, so, at each time point, all participants completed all tests (Figure 1).

### 2.2. Procedures

This was a cohort study with measurements taken at two points in time: September 2023, during the last phase of the pre-season period, and April 2024, at the end of the competition season. At each point, all anthropometric measurements and body composition analysis took place over a period of 2 weeks. Each participant underwent all measurements on the same day, coinciding with a rest day so that physical activity would not affect the scans, under fasting conditions of 8 h, after urinating, and while in underwear or wearing sports shorts.

#### 2.2.1. Anthropometric Assessment

Weight and height were measured using a digital scale (SECA, Hamburg, Germany; accuracy 0.1 kg) and a stadiometer (SECA 213, Hamburg, Germany; accuracy 0.1 cm), respectively. Skinfolds were measured with a Harpenden caliper (Realmet, Barcelona, Spain; accuracy 0.2 mm), circumferences with an anthropometric measuring tape (Realmet, Barcelona, Spain; accuracy 0.1 cm), and diameters with a small anthropometer (Realmet, Barcelona, Spain; accuracy 0.1 cm). The measurements were used to calculate body mass index (BMI) (kg/m^2^), sum of six skinfolds (Sum6F) (triceps, subscapular, supraspinale, abdominal, thigh, calf), and eight skinfolds (Sum8F) (triceps, subscapular, biceps, iliac crest, supraspinale, abdominal, thigh, calf). The Heath and Carter anthropometric method was used for somatotype determination, calculating each of its three components (endomorphy, mesomorphy, and ectomorphy) with their corresponding formulas [13].

Anthropometric assessment was performed in accordance with the International Society for the Advancement of Kinanthropometry (ISAK) restricted profile protocol, which consists of 21 measurements (4 basic measures [kg, cm], 8 skinfolds [mm], 6 girths [cm], and 3 breadths [cm]) [15]. All measurements were taken twice, always on the right side. A third measurement was taken if there were anthropometric technical errors of measurement above the recommended limits: >5% for skinfolds and >1% for all other measurements. The value chosen was the mean in the case of two measurements and the median in the case of three measurements. All measurements were performed by two ISAK Level II trained and accredited professionals; one of them carried out the measurements and the other supervised the procedure and acted as note-taker, always operating in the same position.

#### 2.2.2. Dual-Energy X-Ray Absorptiometry

Body composition was estimated using the GE Lunar Prodigy DXA device (GE Healthcare, Madison, WI, USA; Encore software version 17). The device was calibrated daily with a phantom according to the manufacturer’s guidelines. All subjects were asked to remove any metal objects prior to scanning. Whole-body scans were performed using NHANES positioning criteria. If participants were broader than the scan area, two separate scans were performed to assess the whole body [16]. A single trained DXA technician positioned participants and performed the scans.

The variables considered were total mass (TM; kg), lean mass (LM; kg), fat-free mass (FFM; kg), body fat (BF; kg and %), bone mass (BM; kg), and BMD (g/cm^2^).

### 2.3. Statistical Analysis

Data were expressed as mean (standard deviation). For data analysis, participants were divided into two groups based on their playing position: backs and forwards. Statistically based, grouping players is justified to ensure sufficient statistical power to perform robust comparative analyses [6]. Normality was assessed using the Shapiro-Wilk test. A paired *t*-test was used to assess whether changes occurred before and after the season (for all players and for each group). In case of non-normal distribution, Wilcoxon signed-rank test was applied. Student’s *t*-test was used to determine whether there were differences between positions in the pre- or post-season, or in the change of each variable. If the variables did not follow a normal distribution, the Mann-Whitney U test was used. In the case of heterogeneity, which was checked by Levene’s test, Welch’s test was applied. The effect size of standardized differences was determined using Cohen’s d statistic, and its magnitude can be interpreted using the Hopkins’ scale, where 0 to 0.2 = trivial, 0.2 to 0.6 = small, 0.6 to 1.2 = moderate, 1.2 to 2 = large, and >2 = very large [17]. Significance was set at *p* < 0.05.

All statistical tests were carried out using Statistical Package for the Social Sciences version 23 (IBM, Chicago, IL, USA). All statistical analyses were performed using JASP Team (2024) (version 0.18.3) and Microsoft Excel software (Microsoft, Redmond, WA, USA).

A sensitivity analysis using G*Power 3.1.9.7 indicated that, with *n* = 16 paired observations per positional group (forwards and backs), the study had 80% power to detect within-group pre–post changes of Cohen’s dz ≥ 0.75 (α = 0.05, two-tailed), corresponding to moderate-to-large effects. Accordingly, the study was adequately powered to identify substantial within-group changes across the season, while smaller effects may not have been reliably detectable. Non-parametric alternatives (Wilcoxon signed-rank tests) are expected to require similar or slightly larger true effects to achieve the same level of power [18].

## 3. Results

The characteristics of the sample in terms of age, weight, height, BMI, and years of experience at this competition level are shown in Table 1. At baseline, forwards had significantly higher weight, height, and BMI than backs (*p* < 0.001). There was no significant difference in the age or in the years of experience between backs and forwards.

Looking at the changes in body composition separately by position, back or forward, back players significantly increased TM (*p* = 0.012), LM (*p* = 0.005), FFM (*p* = 0.004), and mesomorph (*p* = 0.008) and decreased ectomorph (*p* = 0.022) over the season, while forward players only increased BM (*p* = 0.031). Additionally, when comparing body composition variables at the beginning and at the end of the season between positions, all evaluated body components were significantly higher in the forwards both before and at the end of the season (Table 2). The changes through the season, without separating by position, showed noticeable changes in TM, LM, FFM, and BM in both backs and forwards. These data are shown in Appendix A.

When analyzing seasonal changes in body composition by playing position, no significant differences were found between backs and forwards, except for ectomorphy, which significantly decreased in backs (−0.138 ± 0.22 vs. 0.04 ± 0.17; *p* = 0.029).

We also compared the changes over the course of the season between the front row and the other forwards and found that the change experienced in LM and FFM was significantly higher in front rows than in the rest of forwards (*p* < 0.05). The rest of body composition changes did not differ between forward players (*p* > 0.05) (Table 3). In Appendix A, the difference between the front rows and other forwards can be observed through the season.

The changes in somatotype are shown in Figure 2, where it was observed that backs changed their position in the *Y*-axis (*p* = 0.003; Cohen’s d = −0.868), but not in the *X*-axis (*p* = 0.069; Cohen’s d = 0.489), and forwards did not suffer any change (*Y*-axis: *p* = 0.509, Cohen’s d = 0.169; *X*-axis: *p* = 0.823, Cohen’s D = 0.057). The change in position in the *Y*-axis was significantly higher for backs compared to forwards (*p* = 0.007; Cohen’s d = 1.033). The change in *X*-axis reported no differences between positions (*p* = 0.526; Cohen’s d = −0.227).

## 4. Discussion

The present study aimed to assess body composition of elite rugby players by position and compare the change in body composition over the season between backs and forwards.

We found significant differences between backs and forwards at baseline in all the measurements made, with forwards having more weight, height, BF, LM, FFM, BM, and BMD. These differences were maintained at the end of the season. This result was expected based on other samples described in the literature since the characteristics of the game demand these differences in body composition between the different positions [4,6,7]. Over the years, alongside with the professionalization of this sport team, there has been an increase in the size of players, both backs and forwards [3,19]. The game has become more physically demanding so the number of collisions has increased considerably [4,11]. The game statistics are evidence of this; over the last 20 years, according to data obtained from the Rugby World Cup (RWC), the average number of collision events in RU per match has risen from 186 in 2003 to 263 in 2023 (scrums, tackles, and rucks included) [20].

Looking at the differences over time in the same position, backs increased significantly TM, LM, and FFM. On the other hand, forwards showed a significant increase in BM. It is unclear whether their high BM is genetically determined or a result of training, though likely a combination of both. In both backs and forwards, players’ ages align with peak BM development in men [21]. However, with regard to the increase in BM experienced by forwards throughout the season, it is conceivable that this could be related to greater number of impacts suffered by these players, acting as a mechanical stimulus for bone formation [22,23]. Some studies have reported twice as many collisions in forwards as in backs (0.7–0.9 collisions per minute vs. 0.3–0.4) [24].

In terms of somatotype, backs were less endomorphic and mesomorphic than forwards but more ectomorphic, as has been observed in other studies with Argentines [6], Spanish [25], and American players [26]. However, changes in body composition over the season were reflected in a change in somatotype, with mesomorphy increasing significantly and ectomorphy decreasing for backs. Then, forwards did not change their somatotype. However, when we compare the change in ectomorphy between the two groups, a significant difference was observed related to the increase in TM at the expense of FFM and LM experienced by the backs.

It is well known that players occupying the front row are the heaviest players of the team [6,27], so we analyzed whether there were any differences between these players and the rest of the forwards. Differences were found in TM at the end of season and pre- and post-season BF evaluated by both DXA and sum of skinfolds. These values were significantly higher in the front row in the same way as has already been seen in Italian players [28]. With respect to somatotype, significant differences have been observed in the three components, both at the beginning and at the end of the season, between front rows and other forwards: front rows are much more endomorphic, slightly more mesomorphic, and noticeably less ectomorphic than other forwards.

Within the group of forwards, in terms of evolution throughout the season, significant differences were observed in LM and FFM. While the front rows increased them, the other forwards slightly decreased them. A study on TM as a performance determinant in RWC players found that forwards from the top teams [winners, finalists, and semi-finalists] were significantly heavier than those from the other teams. Since LM enhances speed, strength, and power [5], FFM may serve as a predictor of success [28].

The lack of a comprehensive and up-to-date database of European RU players makes it very difficult to compare our data. Nevertheless, comparing our sample with another similar sample in age and experience of elite professional Italian players [28], we have observed similar data but slightly lower values in height, TM, and FFM for Spanish both forwards and backs. In the same way, comparing with other nationalities in the southern hemisphere like Australia [11] or New Zealand [7], both measured by DXA, a much larger difference has been observed in height, TM, and LM. In addition, the percentage of BF is lower in these Italian forwards (19.1% vs. 23.17%) and backs (12.4% vs. 14.87%), as well as Australian forwards (14.2% vs. 23.17%) and backs (10.7% vs. 14.87%) than in Spanish players. It is striking that despite the fact that the Sum8SF is lower in New Zealand backs (56.2 mm vs. 71.42 mm), the %BF is extremely similar (14.8% vs. 14.87%); this must be due to the difference in lean mass being much higher in New Zealanders (66.51 vs. 78.5 kg) [7]. BF benefits rugby players by increasing TM, which enhances linear momentum and provides a physical buffer in collisions. However, excessive non-functional fat mass can impair acceleration and thermoregulation, and raise energy cost of exercise [23,28]. In this same New Zealand population [7], we found great similarities with BM in the backs (3.6 kg vs. 3.60 kg), and higher values in Spanish forwards (4.1 kg vs. 4.39 kg), with BMD being higher in Spanish players, both backs and forwards. Although these teams compete in the same category (their country’s premier league), it should be noted that the level of play in Spain is not the same as in countries where rugby is very popular and has a long tradition and professionalism, such as Australia or New Zealand. Thus, as seen in other studies comparing different levels of play [26,28], we agree that the lower the level of play, the lighter, shorter, fatter, and with less fat-free mass the players will be. However, the differences between playing positions are maintained regardless of the level, although the lower levels show greater variability within each position.

Beyond size, collective experience is also a key performance factor. Forwards rely on coordination and teamwork in game elements like the scrum and line-out, making veteran status particularly advantageous for them [5,29,30]. Furthermore, teams winning the RWC from 1987 to 2007 have forwards with a collective experience significantly higher than the teams that did not win [5]. On the other hand, in this study, no differences were observed between backs and forwards, neither in age nor in years of experience playing rugby.

One of the limitations of this study is the grouping into backs and forwards. Due to the heterogeneity of the sample, ideally, a larger sample would allow for the study of positions individually. Within the forwards, there may be differences between specific positions; for example, second-row players may be the tallest on the team due to their role in the lineout. Similarly, within the backs, the halfback may benefit from being short to enhance agility. It would also be interesting to study the body composition by ethnicity [9,11,23], since height and bone mass are parameters limited by genetics; it was observed that Polynesian rugby players appear to have a higher proportion of fat mass in the periphery and a lower proportion in the trunk compared to Caucasian players [11]. Studying the distribution of LM and BF as a performance factor would be valuable. Differences in body distribution between ethnic groups may influence the power-to-mass ratio, enhancing power generation in explosive movements (tackles, mauls, scrums, rucks, etc.) [11,23].

To the authors’ knowledge, this is the first study to examine Spanish RU elite players with DXA in our country. It is worth highlighting that the participants are professional players of the highest level in Spanish rugby, a largely unstudied population, as well as the use of DXA for the measurement of body composition, the most accurate, valid, and reliable technique currently available for assessment.

## 5. Conclusions

Rugby players show differences in body composition based on the physical demands of their playing position. When analyzing changes within each position over time, backs showed significant increases in TM, LM, and FFM, while forwards experienced a significant rise in BM. The analysis of body composition by playing positions helps teams develop targeted improvement strategies in physical training and nutrition, as well as optimize player selection, e.g., monitoring BF allows for maintaining optimal levels that do not compromise speed, agility, and endurance [12]. Similarly, assessing FFM helps to test whether a hypertrophy training and nutrition strategy is adequate, or to detect whether muscle mass is being lost, a risk factor for injuries. The combination of these two measurement methods provides a more complete picture of body composition. Since DXA is more expensive, it could be used for specific measurements that serve as a reference, while anthropometry, which is much more accessible, could be part of routine monitoring. Based on positional differences observed throughout the season, this study suggests that the high number of collisions experienced by forwards may contribute to their increase in BM. Future research on body composition, providing position-specific information and specific distribution by compartments, is needed.

## Figures and Tables

**Figure 1 jfmk-10-00357-f001:**
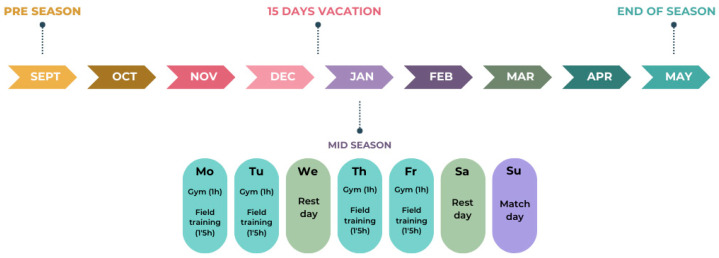
Training outline for participants.

**Figure 2 jfmk-10-00357-f002:**
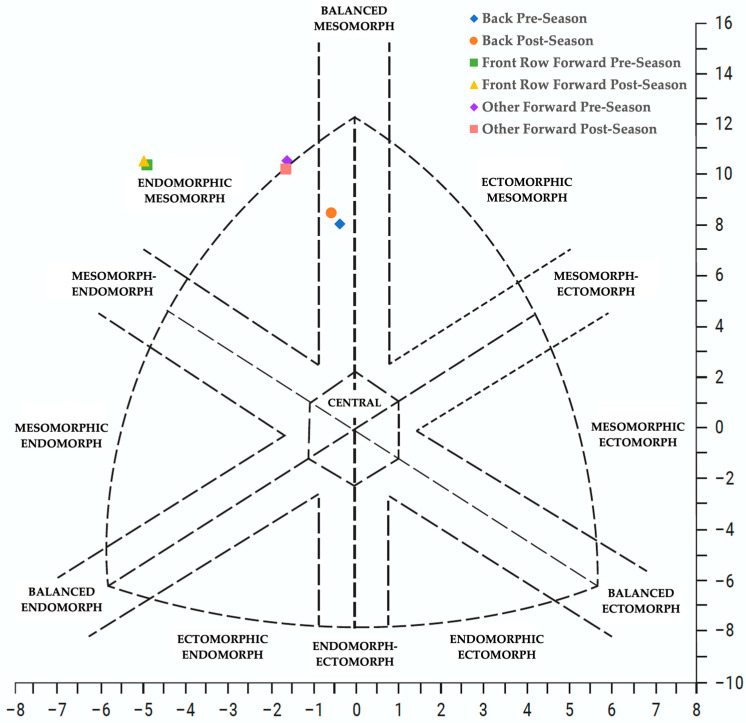
Changes in somatotype during the season according to player position. The data points were calculated using the following equations: *X*-axis = ectomorphy − endomorphy; *Y*-axis = 2 × mesomorphy − (endomorphy + ectomorphy) [13].

**Table 1 jfmk-10-00357-t001:** Characteristics of the participants (*n* = 32).

	Total (*n* = 32)	Backs (*n* = 16)	Forwards (*n* = 16)	*p*-Value	Effect Size (Cohen’s d)
Age (years)	25.97 (4.51)	25.19 (5.01)	26.75 (3.96)	0.335 ^a^	−0.346
Weight (kg)	93.00 (15.39)	80.14 (6.09)	105.84 (10.00)	<0.001 ^a^	−3.105
Height (cm)	181.77 (6.27)	178.20 (4.98)	185.33 (5.42)	<0.001 ^a^	−1.370
BMI (kg/m^2^)	28.03 (3.62)	25.21 (0.95)	30.85 (3.02)	<0.001 ^b^	−2.516
Experience (years)	5.38 (3.93)	5.13 (4.30)	5.63 (3.65)	0.725 ^a^	−0.125

Data are expressed as mean (standard deviation). ^a^ Student’s *t*-test, ^b^ Welch *t* test. Abbreviations: BMI (body mass index).

**Table 2 jfmk-10-00357-t002:** Pre-season (PS) and end-of-season (ES) DXA and anthropometric body composition values in relation to position, differences in each position throughout the season, and differences between positions in pre- and end-of-season values.

Body Composition Variables	Moment of the Season	Backs (*n* = 16)			Forwards (*n* = 16)			Differences Between Backs and Forwards	
		Mean (SD)	*p*-Value (Intra-Group)	Effect Size (Cohen’s d)	Mean (SD)	*p*-Value (Intra-Group)	Effect Size (Cohen’s d)	*p*-Value (Inter-Group)	Effect Size (Cohen’s d)
DXA									
TM (kg)	PS	80.44 (6.12)	0.012 ^a^	−0.710	106.07 (9.13)	0.512 ^a^	−0.168	<0.001 ^c^	−3.296
	ES	81.72 (5.12)			106.81 (9.92)			<0.001 ^d^	−3.126
LM (kg)	PS	65.40 (5.32)	0.005 ^a^	−0.820	78.02 (5.90)	0.344 ^a^	−0.244	<0.001 ^c^	−2.245
	ES	66.51 (4.92)			78.38 (6.07)			<0.001 ^c^	−2.150
FFM (kg)	PS	68.97 (5.58)	0.004 ^a^	−0.838	82.37 (6.29)	0.306 ^a^	−0.265	<0.001 ^c^	−2.255
	ES	70.11 (5.16)			82.77 (6.46)			<0.001 ^c^	−2.165
BF (kg)	PS	11.47 (1.90)	0.638 ^a^	−0.120	23.70 (7.15)	0.730 ^a^	−0.088	<0.001 ^c^	−2.338
	ES	11.60 (1.75)			24.04 (7.66)			<0.001 ^c^	−2.237
BF (%)	PS	14.92 (2.28)	0.872 ^a^	0.041	23.24 (5.67)	0.852 ^a^	−0.047	<0.001 ^c^	−1.879
	ES	14.87 (2.10)			23.17 (5.71)			<0.001 ^c^	−1.930
BM (kg)	PS	3.57 (0.33)	0.066 ^a^	−0.495	4.36 (0.47)	0.031 ^a^	−0.594	<0.001 ^c^	−1.936
	ES	3.60 (0.31)			4.39 (0.46)			<0.001 ^c^	−2.011
BMD (kg/cm^2^)	PS	1.44 (0.09)	0.273 ^a^	−0.284	1.62 (0.11)	0.535 ^a^	0.159	<0.001 ^c^	−1.750
	ES	1.45 (0.08)			1.61 (0.10)			<0.001 ^c^	−1.759
Anthropometry									
Sum6SF (mm)	PS	51.64 (8.11)	0.476 ^a^	−0.183	99.16 (32.44)	0.470 ^a^	−0.185	<0.001 ^d^	−2.010
	ES	52.68 (9.23)			103.06 (33.46)			<0.001 ^d^	−2.050
Sum8SF (mm)	PS	68.16 (11.19)	0.135 ^a^	−0.395	132.56 (44.21)	0.321 ^a^	−0.257	<0.001 ^d^	−1.997
	ES	71.42 (12.58)			141.61 (48.93)			<0.001 ^d^	−1.965
Endomorphy	PS	2.06 (0.40)	0.302 ^a^	−0.267	3.89 (1.45)	0.623 ^a^	−0.125	<0.001 ^d^	−1.731
	ES	2.12 (0.41)			3.99 (1.42)			<0.001 ^d^	−1.776
Mesomorphy	PS	5.88 (0.51)	0.008 ^a^	−0.767	7.48 (0.67)	0.932 ^b^	0.033	<0.001 ^c^	−2.686
	ES	6.07 (0.53)			7.51 (0.78)			<0.001 ^c^	−2.152
Ectomorphy	PS	1.70 (0.38)	0.022 ^a^	0.638	0.65 (0.54)	0.507 ^b^	−0.255	<0.001 ^c^	2.252
	ES	1.56 (0.38)			0.69 (0.57)			<0.001 ^c^	1.808

Data are expressed as mean (standard deviation). ^a^ Paired *t*-test, ^b^ Wilcoxon sign rank test, ^c^ Student *t*-test, ^d^ Welch *t* test. Abbreviatures: BF (body fat), BM (bone mass), ES (end-of-season) FFM (fat-free mass), LM (lean mass), TM (total mass), PS (pre-season), Sum6SF (sum of six site skinfolds), Sum8SF (sum of eight site skinfolds).

**Table 3 jfmk-10-00357-t003:** Seasonal changes in DXA and anthropometric body composition considering forwards’ positions: front rows (FR) and other forwards (OF).

	Change FR	Change OF	*p*-Value	Effect Size (Cohen’s d)
DXA				
TM (kg)	3.061 (4.548)	−1.071 (3.482)	0.344 ^a^	0.490
LM (kg)	1.42 (1.42)	−0.46 (0.92)	0.009 ^a^	1.507
FFM (kg)	1.464 (1.402)	−0.437 (0.951)	0.009 ^a^	1.514
BF (kg)	1.597 (4.362)	−0.634 (3.404)	0.869 ^a^	0.084
BM (kg)	0.040 (0.054)	0.026 (0.060)	0.651 ^a^	0.231
BMD (kg/cm^2^)	−0.003 (0.039)	−0.011 (0.054)	0.564 ^a^	0.295
Anthropometry				
Sum6SF (mm)	13.157 (21.116)	−3.400 (18.341)	0.407 ^a^	0.188
Sum8SF (mm)	28.543 (36.566)	−6.111 (27.178)	0.713 ^a^	0.428
Endomorphy	0.357 (0.793)	−0.111 (0.685)	0.975 ^a^	0.016
Mesomorphy	0.214 (0.406)	−0.122 (0.239)	0.222 ^c^	0.648
Ectomorphy	−0.029 (0.049)	0.100 (0.212)	0.828 ^b^	−0.078

Data are expressed as mean (standard deviation). ^a^ Student *t*-test, ^b^ Mann-Whitney U test, ^c^ Welch *t* test. Abbreviations: BF (body fat), BM (bone mass), FFM (fat-free mass), LM (lean mass), TM (total mass), Sum6SF (sum of six site skinfolds), and Sum8SF (sum of eight site skinfolds).

## Data Availability

The data presented in this study are available on request from the corresponding author due to privacy concerns.

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
