# Peer review of "Seasonal Body Composition Changes in Elite Rugby Players: DXA and Anthropometry-Based Comparison of Backs and Forwards"

_jfmk, 2025, doi:10.3390/jfmk10030357_

Round 1

Reviewer 1 Report

Comments and Suggestions for Authors

Dear Authors,

Thank you for the opportunity to review this manuscript. The manuscript submitted for review focuses on evaluating changes in the basic somatic parameters of rugby players over a periodized training cycle, with consideration for the specificity of their field position. As the authors suggest, the findings may have applicational significance for increasing team success.

The article requires substantial revisions. My comments and suggestions for the authors are as follows, organized by the manuscript's sections:

  1. Keywords: Keywords must be different from the words used in the title. The entire title already functions as keywords, so repetition is unnecessary.

  2. Aim of the study: Please describe the aim of the study more precisely. Expand on what assumptions are to be verified and what the authors' research hypothesis is.

  3. Ethical Approval: Please specify the number of the ethical approval for the research and who issued it in the manuscript text.

  4. Helsinki Declaration: Supplement the manuscript with references to the Declaration of Helsinki.

  5. Materials and Methods: This chapter requires further clarification. Please state in the manuscript whether the sample size ensures the expected power of the tests (e.g., in accordance with G-Power)?

  6. Anthropometric Measurements: Please specify exactly how the anthropometric measurements were taken, including the specific anatomical locations, how many times the measurements were performed, and whether an average was calculated. Also, state on which side of the body the measurements were taken. Providing these details is crucial for other researchers to replicate the study and compare results. Other publications containing anthropometric measurements in athletes describe these issues in great detail (https://doi.org/10.3390/app15148020)

  7. Inclusion/Exclusion Criteria: What were the inclusion and exclusion criteria for the study? Were injuries a factor? Did all invited participants take part in the study? If not, how many declined, why, and are the characteristics of these individuals known? Were all participants who were tested before the season also tested after the season?

  8. Somatic Assessment Methods: The authors have described their choice of two methods for assessing somatic build too superficially. Did one method prove to be more reliable?

  9. Table 2: Table 2 is redundant. The authors should present the results separately for backs and forwards throughout the paper.

  10. Theoretical Chapters: The theoretical chapters require a more insightful description. The manuscript lacks a discussion of the practical, applied conclusions.

Kind regards

reviewer

Reviewer 2 Report

Comments and Suggestions for Authors

Introduction: 
The problem needs to be better defined. On the one hand there is ‘lack of data in amateurs’ and, at the same time, "prioritisation of size as a determinant of performance by coaches.  The central problem could be better defined and accompanied by a consequence of this.

Insufficient justification. It is necessary to explain better why the study, either for impact on individual training planning, according to playing position, why it helps in talent selection, bone health and performance. In addition to why it is necessary to evaluate throughout the season.

In terms of instruments and techniques, explain why to use DXA + anthropometry. Clarify what each one provides and why use them together. 
Make it clear that they do not detect the same thing, they are complementary.

Method
In the introduction, the problem is posed by the lack of data on amateur sportsmen and women, but the sample is made up of professional sportsmen and women, first division honour, is this professional or not? 
Nationality is irrelevant. More important are the calibration of the sample, weight and height were missing. Code of ethics committee, from which university?

Statistics: Reporting effect sizes would give much more practical clarity than just p-values.

Conclusion: 
There was no mention of seasonal differences. 
It would be good to add, as projections and practical application, the importance of seasonal monitoring, the use of combined DXA/Anthropometry techniques and thus have a more complete view of the body composition of athletes that allows technical teams to make better decisions in terms of training and nutrition programmes.
